# PEP4LEP study protocol: integrated skin screening and SDR-PEP administration for leprosy prevention: comparing the effectiveness and feasibility of a community-based intervention to a health centre-based intervention in Ethiopia, Mozambique and Tanzania

Anne Schoenmakers [1], Thomas Hambridge [2], Robin van Wijk [1], Christa Kasang [3], Jan Hendrik Richardus [2], Kidist Bobosha [4], Fernando Mitano [5,6], Stephen E Mshana [7], Ephrem Mamo [4], Abdoulaye Marega [5], Nelly Mwageni [7], Taye Letta [8], Artur Manuel Muloliwa [5,6], Deus Vedastus Kamara [9], Ahmed Mohammed Eman [10], Litos Raimundo [11], Blasdus Njako [12], Liesbeth Mieras [1]

AS and TH contributed equally.

For numbered affiliations see end of article.

**Correspondence to**
Thomas Hambridge;
t.hambridge@erasmusmc.nl

## ABSTRACT

**Introduction** Leprosy, or Hansen's disease, remains a cause of preventable disability. Early detection, treatment and prevention are key to reducing transmission. Post-exposure prophylaxis with single-dose rifampicin (SDR-PEP) reduces the risk of developing leprosy when administered to screened contacts of patients. This has been adopted in the WHO leprosy guidelines. The PEP4LEP study aims to determine the most effective and feasible method of screening people at risk of developing leprosy and administering chemoprophylaxis to contribute to interrupting transmission.

**Methods and analysis** PEP4LEP is a cluster-randomised implementation trial comparing two interventions of integrated skin screening combined with SDR-PEP distribution to contacts of patients with leprosy in Ethiopia, Mozambique and Tanzania. One intervention is community-based, using skin camps to screen approximately 100 community contacts per leprosy patient, and to administer SDR-PEP when eligible. The other intervention is health centre-based, inviting household contacts of leprosy patients to be screened in a local health centre and subsequently receive SDR-PEP when eligible. The mobile health (mHealth) tool SkinApp will support health workers' capacity in integrated skin screening. The effectiveness of both interventions will be compared by assessing the rate of patients with leprosy detected and case detection delay in months, as well as feasibility in terms of cost-effectiveness and acceptability.

**Ethics and dissemination** Ethical approval was obtained from the national ethical committees of Ethiopia (MoSHE), Mozambique (CNBS) and Tanzania (NIMR/MoHCDEC). Study results will be published open access

## Strengths and limitations of this study

► In both interventions, a combination of screening contacts and providing post-exposure prophylaxis with single-dose rifampicin will be used according to the WHO's guidelines to reduce the contacts' risk of developing leprosy.

► An integrated skin screening approach will be used in which multiple diseases can be detected and treated at once, overcoming the often negative associations with leprosy.

► The SkinApp will be used as a mobile health tool to support peripheral health workers in recognising and treating signs and symptoms of skin diseases; while innovative and potentially increasing capacity, the accuracy and reproducibility of this tool awaits further investigation.

► Since the epidemiological impact on the new case detection rate will not become apparent within the study duration, the primary outcome measures are case detection delay, number of contacts diagnosed with leprosy and number of contacts who received chemoprophylaxis.

► Because difficulties in recalling the first signs and symptoms are expected to increase over a longer duration of the disease, only recently diagnosed index patients will be included in this study to establish case detection delay.

in peer-reviewed journals, providing evidence for the implementation of innovative leprosy screening methods and chemoprophylaxis to policymakers.

**Trial registration number** NL7294 (NTR7503).

## INTRODUCTION

Leprosy, or Hansen's disease, is a communicable disease caused by *Mycobacterium leprae* that is still a public health problem in many countries. It is formally recognised by the WHO as a neglected tropical disease (NTD).[1] The annual reported number of newly detected patients with leprosy was 202 185 in 2019.[2] If left untreated, leprosy potentially results in disability, which can have severe consequences such as stigma and poverty.[3] Leprosy has a long and variable incubation time, ranging from 2 to 20 years, during which it is assumed that transmission can take place.[4] The risk of developing leprosy is higher in household contacts and neighbours of patients than it is in the general community.[5] Moet *et al* demonstrated that physical and genetic distance were independently associated with the risk of a contact developing leprosy.[6] According to the WHO, contact screening should be offered to a person who has been in contact with an untreated leprosy index case for at least 20 hours per week during at least 3 months in the previous year.[4 7 8] An index case is defined as a person diagnosed with leprosy for the first time.[7]

The WHO has provided multidrug therapy free of charge to all patients with leprosy since 1995.[9] However, to overcome ongoing transmission in high-endemic areas, innovative measures are needed.[8 10] In 2008, a large randomised controlled trial in Bangladesh (Chemoprophylaxis for leprosy study, COLEP) demonstrated that a single dose of rifampicin (SDR) given to contacts of newly diagnosed patients with leprosy is effective in reducing the risk of leprosy by 57% (95% CI: 24% to 75%).[11] Post-exposure prophylaxis with single-dose rifampicin (SDR-PEP) was found to be cost-effective in Bangladesh.[12] In the Leprosy Post-Exposure Prophylaxis (LPEP) programme, SDR-PEP was implemented in areas representing various health systems across three continents and eight countries, to evaluate the feasibility, effectiveness and impact.[13] The implementation of SDR-PEP within the routine leprosy control programmes was proven to be safe and generally well accepted. Based on the LPEP programme and a microsimulation leprosy model (SIMCOLEP), SDR-PEP was also found to be cost-effective in India.[14] The concern that SDR-PEP could lead to increased rifampicin resistance in other diseases, such as tuberculosis (TB), was considered in an expert consultation that concluded that SDR-PEP given to contacts of patients with leprosy, in the absence of symptoms of active TB, poses a negligible risk of generating resistance in *M. tuberculosis* in individuals and in populations.[15] In 2018, SDR-PEP was included in the WHO 'Guidelines for the diagnosis, treatment and prevention of leprosy'. Once contact tracing has been established, SDR-PEP can be included into the routines of leprosy control programmes with minimal additional efforts and costs.[7 16]

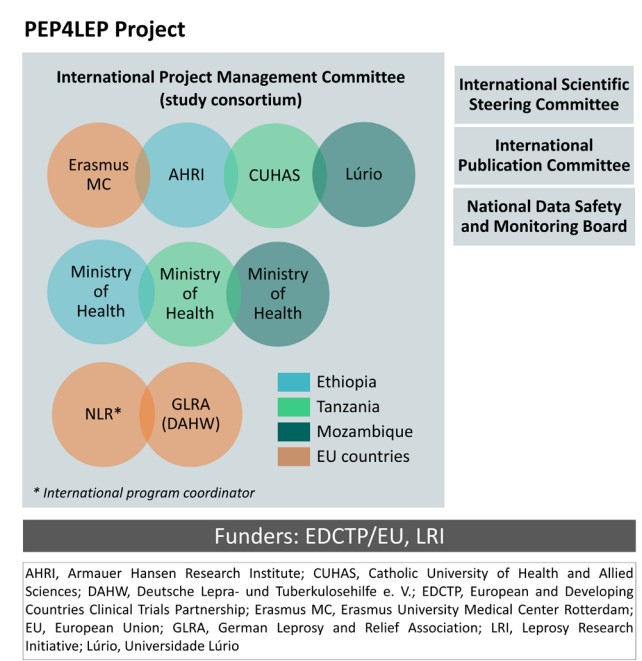

**Figure 1** PEP4LEP project organisation chart.

Skin screening is an important detection strategy for skin-NTDs such as leprosy, and is recommended to be embedded in leprosy programmes.[1 7 17 18] Screening for multiple skin diseases at once (integrated or common skin screening) is promoted by WHO.[1 8 19 20] Integration is considered to increase effectiveness and efficiency by minimising costs and expanding intervention coverage.[19 21] An important obstacle for integrated skin screening is the scarcity of dermatologists in many areas with a high skin NTD endemicity.[22] In sub-Saharan Africa, the situation is critical, with approximately one dermatologist per 500 000–1 million inhabitants and even larger shortages in Mozambique and Tanzania according to oral field reports from PEP4LEP consortium members.[23 24] According to the WHO, community health workers and village volunteers can play a role in screening for skin diseases, but improved knowledge, capacity and motivation of health workers and community volunteers is essential.[17 19 25–29] As both integrated skin screening for NTDs and SDR-PEP against leprosy are promoted by the WHO, additional implementation studies are necessary to establish whether a combined intervention is acceptable, feasible and cost-effective in leprosy endemic areas.[1 4 8 19]

### Objectives

The PEP4LEP project is a collaboration among study consortium members in five countries in sub-Saharan Africa and the European Union (EU) (figure 1). The overall aim of this cluster-randomised implementation trial is to contribute to the interruption of *M. leprae* transmission by identifying the most effective and feasible method of screening people at risk of developing leprosy and by administering post-exposure chemoprophylaxis in Ethiopia, Mozambique and Tanzania. The primary study

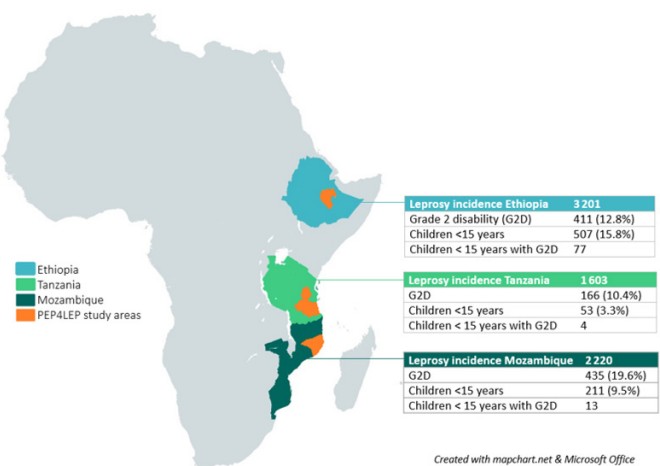

**Figure 2** PEP4LEP countries' leprosy incidence in 2019 according to the WHO (2020).[2]

objectives are to compare the effectiveness and feasibility of a community-based screening and prophylaxis intervention (skin camp) with a health centre-based screening and prophylaxis intervention solely for household contacts of a leprosy patient. The case detection delay will be the primary outcome measure to assess effectiveness. Additional objectives are to assess the cost-effectiveness, acceptability and health workers' capacity regarding the integrated skin diseases approach and the use of the supportive mobile health (mHealth) tool SkinApp.[30 31]

## METHODS AND ANALYSIS
### Study setting
This study will take place in three countries in sub-Saharan Africa: Ethiopia, Mozambique and Tanzania. The three countries differ socioculturally and in the endemicity of NTDs like leprosy (figure 2).[2] Districts within these countries were purposefully chosen because of endemicity and the focal distribution of reported leprosy cases. In Ethiopia, three endemic districts are located in East Hararghe Zone (Oromiya region): Girawa, Jarso and Midega. In Mozambique, the included districts are located in Nampula province: Meconta, Mogovolas, and Murrupula. The Tanzanian districts are Lindi in Lindi Region and Morogoro and Mvomero in Morogoro Region. The original overall study period was October 2018 until January 2023, with an estimated duration of 2.5–3 years for the inclusion of patients with leprosy and their contacts. A study extension is expected due to the impact of COVID-19.

### Participants and eligibility criteria
Patients with leprosy enrolled in the PEP4LEP study are referred to as 'index patients'. These patients were derived from the leprosy programme registries, and preferably diagnosed up to 6 months prior to inclusion to prevent recall problems when assessing the delay in case detection.[32] The inclusion and exclusion criteria for index patients and contacts are summarised in

table 1 and are based on the WHO guidelines and the LPEP programme.[4 13] Following the emergence of the COVID-19 pandemic, a suspicion of a COVID-19 infection was added as contact exclusion criteria for this study, as physical distancing cannot be guarded when performing skin screening.[33–36] Study participants recovered from COVID-19 can still be included after they have been tested negative and are symptom-free for at least 2 weeks.[33–35]

The target population for the feasibility component of this study as well as the other research objectives, consists of various stakeholders, including: (index) patients, household contacts, community contacts, community leaders, health workers, community health volunteers and health policy decision-makers. If applicable, contacts refusing to receive skin screening and/or SDR-PEP, but who are willing to participate in the qualitative component will also be included in the project, contributing to the acceptability component of the study.

The exclusion criterion for these stakeholders is refusal to provide informed consent to participate.

### Study design
The study is a two-arm, cluster-randomised implementation trial (figure 3). One intervention is community-based, using skin camps to screen approximately 100 community contacts (household members and neighbours) of an index patient with leprosy and to provide them with SDR-PEP when eligible. The second intervention is health centre-based, inviting the household contacts of an index patient to be screened and given SDR-PEP when eligible.

### Community-based skin camp intervention
A skin camp will be organised when a patient with leprosy is diagnosed by inviting approximately 100 people from the same community (table 1) living in the surrounding area (field definition: inhabitants from the 20 closest houses). Community contacts from outside of the 20 closest households who attend a skin camp can still receive skin screening or referral, but will not be given SDR-PEP. Health camps are designed to bring specialised care closer to the community, thus expanding healthcare access.[37] Besides providing preventive and curative treatment, these camps often also play a significant role to create awareness.[38] Community 'skin health camps' have been proposed as an effective way to screen for leprosy and other NTDs.[7 39] Skin camps are organised at the community level and in close collaboration with community leaders and local organisations.[37 40] In a skin camp, health staff screen individuals for skin diseases and then treat or refer patients if necessary. Assistance from a dermatologist (or, if none available, a senior health staff member with sufficient dermatology experience) is vital.[41] A key advantage of this community intervention is that the identity of the person affected by leprosy can be protected since no individual disease disclosure is needed. This non-disclosure approach is of utmost importance, as people affected by leprosy are often stigmatised

**Table 1** PEP4LEP eligibility criteria patients and contacts[4 7 13]

| | Index patients | Contacts |
|---|---|---|
| Inclusion criteria | ► Consent to participate in the PEP4LEP project.<br>► Diagnosed with leprosy (preferred maximum of 6 months prior to inclusion).[32]<br>► Residence in the PEP4LEP districts for ≥3 months prior to the date of diagnosis.<br>► Index patient has started MDT.<br>► Community-based skin camp intervention: Patient with leprosy gives permission for the set-up of a skin camp in his/her community (sharing their leprosy diagnosis with their contacts is not needed).<br>► Health centre-based household screening intervention: Patient with leprosy with household contacts, and who is willing to inform these contacts about PEP4LEP. | ► Consent to participate in the PEP4LEP project.<br>► Community-based skin camp intervention: Community contact (living in the 20 closest houses to the index-patient) for ≥3 months.<br>► Health centre-based household screening intervention: Contact which is a household member of the index patient for ≥3 months, visiting the screening health centre ≤3 months after the index patient was included. |
| Exclusion criteria | ► Index patient or parents/legal guardians unable to understand the purpose and risks of participating in the PEP4LEP study. | ► Contact or parents/legal guardians unable to understand the purpose and risks of participating in the PEP4LEP study.<br>► Age <2 years and/or <10 kg of weight.*<br>► Pregnancy.*<br>► Receiving or having received rifampicin for any reason in the last 2 years.<br>► Known allergy to rifampicin.<br>► History of liver or renal disorders.<br>► Individuals with leprosy and those who have possible signs and/or symptoms of leprosy (eg, leprosy-like skin lesions or nerve manifestations) until their disease status has been clarified.†<br>► Individuals with possible signs and/or symptoms of TB (cough for more than 2 weeks or cough in known patients with HIV/AIDS, night sweats, unexplained fever, weight loss) until their disease status has been clarified.‡[113]<br>► Individuals with possible signs and/or symptoms of COVID-19 (self-assessed temperature ≥38°C, respiratory or cold-like symptoms, sudden loss of smell/taste) or possible contact with a patient with COVID-19 in the past 14 days.‡[33–36] |

*A voucher will be given for repeated skin screening and SDR-PEP. This can be used in a PEP4LEP affiliated health centre when this person becomes eligible (eg, after giving birth).
†If referral was needed and no leprosy is detected, repeated skin screening and SDR-PEP can be provided in a PEP4LEP affiliated health centre.
‡Skin screening and SDR-PEP can only be provided in a PEP4LEP affiliated health centre after the contact is tested negative for COVID-19/TB (according to national guidelines).[33–36]
MDT, multidrug therapy; SDR-PEP, single-dose rifampicin post-exposure prophylaxis; TB, tuberculosis.

and discriminated against and are therefore reluctant to share their disease status.[42–44] Moreover, including a wider group of contacts and using an integrated skin diseases approach may overcome the frequently negative associations with leprosy that can prevent people from participating in a leprosy-related intervention.[19] Including approximately 100 contacts per identified patient with leprosy in the PEP4LEP skin camps is in-line with the risk profiles of the contact groups and is operationally manageable to conduct within 1–2 days, also when using time slots to prevent crowding, taking COVID-19 into consideration.[6 13 34 36 38 39 45–47]

### Health centre-based intervention for household contacts

In the second intervention, newly detected patients with leprosy will be asked to invite their household contacts to visit a health centre to have their skin screened and, if eligible, to be offered SDR-PEP. Clustering of the disease within households is commonly seen.[6 47 48] Household contacts are defined as living under the same roof as the index patient with leprosy for a minimum of 3 months (table 1).[13 49 50] To prevent re-infection within a household and for operational management reasons, contacts need to visit the health centre within 3 months after the index patient was included, which is also in-line with contact tracing interventions in literature.[51] Around six

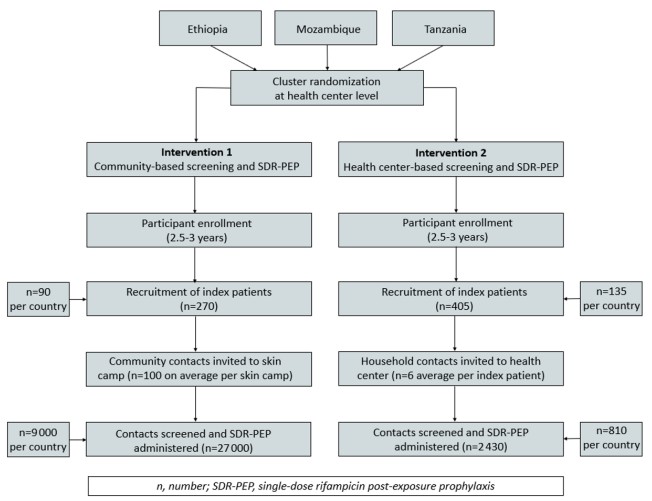

**Figure 3** Flow of participants through the PEP4LEP study.

household contacts per patient are expected to visit the health centre for screening.[13] Previous studies showed that patients with leprosy are usually willing to disclose their leprosy diagnosis to their household members to facilitate screening and prophylaxis, but they are often reluctant to share this information with neighbours or other social contacts.[42–44]

### Integrated skin screening

For contact screening in both interventions, an integrated skin diseases approach—also called common skin screening approach—will be used to diagnose common skin diseases (eg, eczema), skin conditions related to HIV/AIDS (eg, Kaposi's sarcoma) and skin-NTDs (eg, leprosy). 'Integration' in this context refers to combined screening for a minimum of two diseases at the same time in the same communities.[52] In the PEP4LEP project, free topical treatment for the most frequently diagnosed skin diseases will be provided as well as referral advice, in-line with WHO and national medical guidelines.[53–57] The screening for signs and symptoms of skin diseases, as well as the chemoprophylaxis distribution, will follow standard operating procedures (SOPs) in which the eligibility criteria for SDR-PEP are clearly stated. In both interventions, the integrated skin diseases approach will be used and supported by the SkinApp, an mHealth tool developed by NLR and Erasmus University Medical Center (Erasmus MC).[30 31] The SkinApp will support peripheral health workers in recognising and treating signs and symptoms of skin diseases, including skin-NTDs like leprosy.[30 31] A senior health staff member with sufficient dermatology experience (preferably a dermatologist) will attend in person or via secure medical messaging via the application (app) Siilo.[58]

### Post-exposure prophylaxis

Chemoprophylaxis with SDR-PEP has been adopted in the 2018 WHO 'Guidelines for the diagnosis, treatment and prevention of leprosy'.[4] The SDR-PEP dosages used in this project (table 2) are consistent with these WHO guidelines, the 2020 published WHO document 'Leprosy/Hansen disease: Contact tracing and post-exposure prophylaxis. Technical guidance' and the LPEP programme.[4 7 13]

Contacts who are temporarily ineligible to receive SDR-PEP (eg, because of pregnancy, table 1) will receive skin screening and an SDR-PEP voucher, useable in an affiliated PEP4LEP health centre when becoming eligible (eg, after giving birth). Contacts receiving SDR-PEP will also

| Table 2 | PEP4LEP single-dose rifampicin dosages[4 7 13] |
| --- | --- |
| **Age and body weight of contact** | **Rifampicin dosage** |
| ≥15 years | 600 mg |
| 10–14 years | 450 mg |
| 6–9 years and body weight of ≥20 kg | 300 mg |
| ≥2 years old and body weight between 10–20 kg | 150 mg |

receive an SDR-PEP Red Card to keep in their homes. This card indicates that the person has received SDR-PEP for leprosy prevention and is ineligible to receive this again within the next 2 years. These methods were previously used as part of the LPEP programme in Tanzania.[13] In PEP4LEP, serious adverse events (SAEs) will be reported and followed up according to national and PEP4LEP guidelines (see ethical section).[59]

### Outcomes

The primary objectives of this study are to identify the most effective and feasible approach for screening contacts of patients with leprosy in combination with administering chemoprophylaxis to prevent leprosy (table 3). Because of the long incubation period of leprosy, it will not be possible to observe reduced transmission at the population level, in terms of a reduced new case detection rate, during this project period. The active case finding component and raised awareness, however, are expected to lead to more detected cases, improved early case detection and disability rates at time of diagnosis. We hypothesise that enhanced case finding and integrated skin screening will lead to an overall reduction of detection delay primarily in the community-based intervention over the study duration, driven by the diagnosis of patients with early signs of leprosy (and shorter delays) that would otherwise go undetected.

#### Primary outcome measures

The primary outcome measures of effectiveness in the comparison of the two interventions are:
1. Case detection delay, measured in months since the first signs or symptoms of leprosy until diagnosis and in the number of patients with grade 2 disability.
2. Number of new patients with leprosy, subdivided into child proportion, female proportion and multibacillary/paucibacillary classification.
3. Number of contacts screened and receiving SDR-PEP.

#### Secondary outcome measures

Feasibility will be assessed by looking at outcome measures related to cost-effectiveness and acceptability (table 3):
- A cost-effectiveness analysis will be undertaken in the third year of the project, encompassing the costs incurred by the health system and the beneficiaries (out-of-pocket expenditure). It will include collecting indicators such as unit costs, costs per case detected, costs per disability-adjusted life years averted and costs per extra case found. The current practice 'routine service provision' will be compared with the two study interventions.
- The acceptability of both interventions will be determined by comparing the number of index patients and contacts included, as well as by using qualitative research methods, such as semi-structured interviews guided by topic lists, focus group discussions (FGDs) with relevant stakeholders and potentially ethnographic observations during the interventions

**Table 3** PEP4LEP project outcomes and methods of analysis

| Objective | Outcome | Hypothesis | Outcome measure | Method of analysis |
|---|---|---|---|---|
| 1.1 To compare the effectiveness of a skin camp prophylaxis intervention with a health centre-based prophylaxis intervention in terms of the rate of patients with leprosy detected and delay in case detection | Primary: Case detection delay | Reduction in case detection delay is expected to be greater in the community-based intervention compared with the health centre-based household contact approach | Number of months since first signs or symptoms of leprosy until diagnosis (including assessing both 'patient delay' and 'health-system delay'); G2D percentage among newly diagnosed patients with leprosy | Descriptive statistics; multivariate models; non-parametric tests |
| | Primary: Number of contacts diagnosed with leprosy | The community-based intervention will identify more cases of leprosy from contact screening compared with the health centre household contact-based approach | Number of contacts diagnosed with leprosy; child proportion; female proportion; MB/PB classification of newly diagnosed patients with leprosy | Descriptive statistics; Pearson's $\chi^2$ test; Fisher's exact test; multivariate logistic regression analysis |
| | Primary: Number of contacts who received chemoprophylaxis | The community-based intervention will allow more contacts to be screened and receive SDR-PEP compared with the health centre-based household contact approach | Number of contacts screened; number of contacts who received SDR-PEP | Descriptive statistics |
| 1.2 To compare the feasibility of the two chemoprophylaxis interventions (screening household contacts or screening contacts via skin camps) in terms of cost-effectiveness and acceptability | Secondary: Cost-effectiveness of each intervention | The community-based intervention will be more expensive but will have a greater impact compared with the health centre-based household contact approach | Number of index patients included; number of contacts screened; number of cases prevented; number of disabilities avoided; operational costs; out-of-pocket expenses | Health economic evaluations |
| | Secondary: Acceptability of each intervention | Both interventions will be accepted in participating countries | Number of index patients included; number of contacts screened; and qualitative outcomes | Descriptive statistics; qualitative content analysis of interviews; FGDs and potentially observations |
| 2.1 To assess the acceptability of an integrated skin diseases approach and the use of the SkinApp | Additional: Number of contacts diagnosed with other skin diseases | The community-based intervention will identify more cases of other skin diseases from contact screening compared with the health centre-based household contact approach | Number of contacts diagnosed with skin diseases and with NTDs that manifest with skin lesions | Descriptive statistics; Pearson's $\chi^2$ test; Fisher's exact test; multivariate logistic regression analysis |
| | Additional: Acceptability of an integrated skin screening approach and the use of the SkinApp | The integrated skin screening approach will encourage screening participation, and the SkinApp will help health workers to diagnose skin diseases | Number of contacts diagnosed with skin diseases and with NTDs that manifest with skin lesions; utilisation of the SkinApp during contact screening; and qualitative outcomes | Descriptive statistics; sensitivity and specificity; positive and negative predictive values; qualitative content analysis of interviews, FGDs and potentially observations |

 Schoenmakers A, *et al. BMJ Open* 2021;**11**:e046125. doi:10.1136/bmjopen-2020-046125

**Table 3** Continued

| Objective | Outcome | Hypothesis | Outcome measure | Method of analysis |
|---|---|---|---|---|
| 2.2 To compare the capacity of health workers in diagnosing leprosy, other skin diseases and other NTDs that manifest with skin lesions before the start of the study with their capacity in the third year | Additional: Capacity of health workers in diagnosing leprosy and other skin diseases | Participation in training and the use of the SkinApp will improve health worker capacity | Results of health worker capacity assessments and qualitative outcomes | Descriptive statistics; qualitative content analysis of interviews, FGDs and potentially observations |

FGD, focus group discussion; G2D, grade 2 disability; MB, multibacillary; NTD, neglected tropical disease; PB, paucibacillary; SDR-PEP, post-exposure prophylaxis with single-dose rifampicin.

for further data validation. More in-depth (country-specific) protocol descriptions on the acceptability and cost-effectiveness side-studies will be developed together with health economist(s) and social scientist(s).

### Additional objectives

The additional objectives are to assess the acceptability of integrated skin screening and the use of the SkinApp as a supporting mHealth tool in the field, as well as health workers' capacity regarding the integrated skin screening approach (table 3). This will be measured by the number of contacts diagnosed with skin diseases and NTDs and by recording the use of the SkinApp during contact screening. The capacity of health workers to diagnose leprosy and other skin diseases will be determined by a series of four assessments in which the SkinApp can be used: before (baseline) and after PEP4LEP training, during the study and at the end of the study. The four assessments were designed in collaboration with an educational specialist and each include 30 questions (20 multiple choice questions on leprosy and 10 skin disease cases of which five formulated as open questions). The primary PEP4LEP health worker training is conducted over several days and consists of interactive training modules focusing on: the PEP4LEP research project, integrated skin screening including the use of mHealth tools (NLR's SkinApp and Siilo), clinical leprosy and the administration of SDR-PEP.[4 27 28 30 58 60 61] Refresher trainings will also be organised. In addition to the assessments, qualitative methods including semi-structured interviews, FGDs and potentially ethnographic field observations will be used to gain a more in-depth understanding of these objectives.

### Case detection delay

Case detection delay is defined by Muthuvel *et al* as the number of months between the onset of signs and symptoms of leprosy and the time of diagnosis, including both 'patient delay' (period in months between noticing the first sign/symptom to the first healthcare provider consultation) and 'health-system delay' (period in months between the first healthcare provider consultation and the patient receiving the leprosy diagnosis).[62] Several studies have investigated delay in leprosy diagnosis in countries like Bangladesh, Brazil, India, Colombia and Paraguay.[62–69] However, recent literature on delay in diagnosis is limited and mainly focuses on other geographical regions. Therefore, delay will be determined with a structured questionnaire that was designed in the project countries along with input from several stakeholders, which will be shared open access (publication expected). The questionnaire includes two annexes: a set of clinical photos of signs of leprosy and a context-specific calendar indicating important local dates, such as festivities, agricultural seasons and religious celebrations.[70 71] A 'Question-by-Question Guide' was designed to provide support in the administration of the questionnaire. The questionnaires were culturally validated in all three countries, based on the conceptual framework of Herdman *et al* (publication expected).[72]

### Sample size

The sample size calculation was based on case detection delay as the main outcome measure for comparing the effectiveness of each intervention. This measure was used for the calculation because the epidemiological impact (ie, reduction in overall new case detection rate in PEP4LEP districts) will not become apparent within the study duration due to the long incubation time of leprosy. The mean or median delay will be compared between both interventions and with the baseline. A baseline case detection delay will be estimated in each country by interviewing recently diagnosed patients with leprosy with the same structured questionnaire prior to the start of the study. For the sample size calculation, a literature-based estimated average case detection delay of 24 months for patients with leprosy with a SD of 8 months was used, with the conservative assumption that a minimal delay difference of 3 months would be detected between both interventions.[73 74] In order to achieve this, we aim to include at least 675 index patients in the study: 270 in the community-based intervention areas (30 per country

per year) and 405 new patients in the health centre-based intervention areas (45 per country per year). Approximately 100 contacts will be screened per index patient in the community-based intervention areas, and approximately 6 contacts will be screened per index patient in the health centre-based intervention areas; thus, a total of approximately 30 000 contacts will be screened (figure 3). We expect no major differences in case detection delay between clusters and within clusters, hence no significant design effect is foreseen. For the feasibility study component and additional research objectives, interviews and FGDs are planned. For the interviews, a minimum of 10 index patients, 10 household contacts, 10–20 community contacts, 10 health workers/community volunteers, 4 health decision-makers and 10 community leaders will be included. For the FGDs, 6–10 participants will be included: two groups of index patients, two groups of household contacts, two groups of community contacts, two groups with health workers and one to two groups with decision-makers. Contacts refusing to receive skin screening and/or take SDR-PEP, but who are willing to participate in the qualitative study component and community members outside of the 20 closest houses to the index patient in intervention one will also be included. The qualitative research sampling will be purposive, according to the defined target groups, and balanced according to, for example, gender, age, education level, religion and/or socio-cultural background. All fully trained health staff involved in the PEP4LEP project will be asked to consent to enrol in the capacity assessment.

### Randomisation
PEP4LEP used randomisation without blinding at the (clustered) health centre level (health centres/posts), ensuring that clusters were similar in size. There are 17 health facilities included in Ethiopia, 22 in Mozambique and 23 in Tanzania. Blinding is not possible because of the varying operational components of the interventions. Cluster randomisation is commonly used when trying to capture the impact of an intervention at community level on both infectiousness and susceptibility.[75] This method is stated to be feasible logistically, and contamination (eg, information-sharing between participants from both interventions) is unlikely.[75] Randomisation was performed using the statistical software package R.[76] Per country, health centres were randomly divided into the community-based intervention or health centre-based intervention.

### Data collection and management
The PEP4LEP data management plan was developed by Erasmus MC in collaboration with the consortium members. Regarding quantitative data, collectors will record their findings onto paper-based forms. Information from the forms will be entered into the Research Electronic Data Capture (REDCap) system from Vanderbilt University.[77] The REDCap software will be linked to a centralised database server hosted by Erasmus MC.

To determine the cost-effectiveness, data for establishing costs (such as infrastructure, human resources, transportation) and output (such as number of contacts seen, rifampicin capsules provided, patients diagnosed with other NTD-related skin diseases and treatments provided) will be derived from the ongoing surveillance data. Other costs (such as general programme costs, treatment costs and other direct or indirect costs) will be collected from ancillary studies.

Besides quantitative data, qualitative data will also be collected for the acceptability and health workers' capacity assessment. Data from (semi-)structured interviews, FGDs and possible ethnographic observations will be audio-recorded and/or paper-based. Data will be transcribed (verbatim), translated to English and entered into computer-assisted qualitative data analysis software.[78] The transcriptions will be securely stored at Erasmus MC after analysis. A system of identification (ID) codes has been developed to record and maintain data systematically, as well as to maintain 'pseudo-anonymity.'

### Data analysis
Data from the PEP4LEP study will be analysed primarily through quantitative methods using descriptive analysis for all variables (table 3). Mean or median case detection delays will be compared between both interventions and the established baseline. This includes newly diagnosed cases identified through each contact screening intervention as well as those detected through ongoing passive case finding, currently the primary method of detection in routine leprosy programmes in the three countries. The p values for each statistical test will be two-tailed with $p \leq 0.05$ considered significant and 95% CIs presented for regression analyses. Quantitative analysis will be conducted using statistical software such as SPSS.[79]

The acceptability and capacity assessments will include qualitative research data (table 3), which will be coded and analysed using computer-assisted qualitative data analysis software, including Atlas.ti.[78 80] Data coding is necessary to categorise and define what the data signify by identifying concepts, patterns, relations and themes.[81] Data reanalysis will occur until no new topics are emerging and data saturation is reached, which means that no significant new themes are emerging.[82]

### Availability of data and materials
Data will be stored for 25 years according to EU regulation 536/2014 considering clinical medication-related research projects.[66] Data will be made available in a repository for potential authorised reuse for future data analysis or study replication. Sharing data and study materials as well as open access publishing are important values of the EU research and innovation programme Horizon 2020, the European and Developing Countries Clinical Trials Partnership and the PEP4LEP consortium.[66 83] Study materials will be made available via https://www.infolep.org, the international knowledge centre for information resources on leprosy, and via https://www.infontd.org,

**Table 4** PEP4LEP ethical approvals

| Country | Ethical board | Outcome | Primary approval/waiver date |
| --- | --- | --- | --- |
| Ethiopia | National Research Ethics Review Committee from the Ministry of Science and Higher Education (MoSHE) | Approved | 17 February 2020 |
| Mozambique | Comité Nacional de Bioética para a Saúde (CNBS) from the Ministério da Saúde | Approved | 16 August 2019 |
| Tanzania | Ethical Clearance Committee linked to the National Institute for Medical Research (NIMR) and Ministry of Health, Community Development, Gender, Elderly & Children (MoHCDEC) | Approved | 17 June 2019 |
| The Netherlands | Medical Ethics Committee Erasmus University Medical Center Rotterdam (Erasmus MC) | Waiver | 11 April 2019 |

the one-stop portal of information on cross-cutting issues in NTDs.[84 85]

## Patient and public involvement

Community leaders, people affected by leprosy and representatives of disabled people organisations are involved in monitoring the study as well as in mobilising community participation. Results will be reported back to the communities via community workshops. Capacity building is an important part of this project. Besides training health staff and community volunteers, four PhD-candidates will obtain a PhD from this project, of which three candidates originate from the endemic countries included in this project to increase local research capacity.[60]

## Ethics

Ethical approval was obtained in each country according to national guidelines (table 4). Erasmus MC, as European consortium member, received a waiver of full medical ethics review and approval from its ethical board according to the Dutch Medical Research Involving Human Subjects Act (Wet Medisch-Wetenschappelijk Onderzoek met mensen).[86]

Written (or thumbprint) informed consent will be obtained from all study participants. If a participant is below 18 years old, a parent or legal guardian will be asked for consent. Study information given to the study participants prior to asking for consent contains details about: leprosy; the study purpose; the right to withdraw; anonymity of the disease status in the community intervention; the fact that SDR-PEP leads to a leprosy risk reduction and not absolute prevention (ie, awareness of leprosy signs/symptoms remains important after taking SDR-PEP); possible side effects of SDR-PEP (ie, urine discolouration) and adverse events (AEs); the incidental findings procedure; data privacy/safety and national contact information. AEs are expected to be rare after SDR-PEP. In the LPEP study, in which SDR-PEP was administered to 151 928 screened contacts, a single AE was reported (an allergic reaction to rifampicin in Brazil) and no SAEs were seen.[13] Urine discolouration, a common rifampicin side effect, was not considered as an AE requiring follow-up in LPEP. Nevertheless, in (chemo)prophylaxis programmes AEs are of utmost importance because large numbers of healthy individuals are involved. In PEP4LEP, SAEs will be reported following national pharmacovigilance

guidelines and by using the PEP4LEP AE form for registration and to inform the principal investigator.[13 59] The PEP4LEP project's SOP on rifampicin administration therefore included the availability of an emergency allergy kit at community study sites where no health centre is located, which should be used according to national medical/pharmacological guidelines.[55–57] All participants with suspected AEs will be referred for proper medical management and treated free of charge according to national standard treatment guidelines.[59]

Throughout both screening interventions and research projects involving human subjects, incidental findings with potential health importance may be observed.[87] Incidental findings are discoveries made during a research or screening project which are outside the scope of the project.[88] Examples of possible incidental findings when performing full body skin screening include: signs of cancer, venous insufficiency, bleeding diathesis, herniation, dental problems or indications of possible abuse. Incidental findings in a research setting are often not explicit enough to be used for diagnosis, treatment, or clinical care.[89]

The procedures for reporting both SAEs and incidental findings are included in the evidence-based PEP4LEP SOPs, on the participant information sheet and in the health workers' training[59 87 88 90 91] The importance will also be emphasised during ongoing monitoring activities, including field visits.[13]

During the developmental phase of this project, the COVID-19 pandemic emerged. Regarding COVID-19, national governmental and WHO guidelines will be followed.[33–36] Information about COVID-19 and project implications (eg, physical distancing, working in time slots) are included in the project's SOPs, information, education and communication (IEC) materials and health workers' training. Hand washing facilities and personal protective equipment (PPE) such as gloves, face masks and aprons, will be made available at the study sides.

A code of conduct will be designed for the PEP4LEP consortium, based on the code of conducts from WHO and All European Academies.[92 93] All researchers in the project are encouraged to participate in good clinical practice courses, facilitated by the research consortium.[94] National data-safety monitoring boards, an international

publication committee and an international scientific steering committee were formed to monitor the project (figure 1).

## Trial registration

The PEP4LEP project is registered at The Netherlands Trial Register, registration date 10 September 2018.[95]

## Dissemination

Study outcomes are expected to be relevant for other sub-Saharan countries, but also for leprosy endemic areas outside the African context. Results will be shared open access via peer-reviewed journals, at conferences and via infolep and infoNTD.[84 85] Best practices will be shared with the Global Partnership for Zero Leprosy (GPZL).[96] Communities affected and local and national policymakers will be informed on the study outcomes via community meetings/workshops. In addition, project recommendations will be offered to all relevant authorities and the WHO in Ethiopia, Mozambique and Tanzania; the uptake of SDR-PEP into national leprosy guidelines is advised by the WHO.[8]

## DISCUSSION

The PEP4LEP study will use an integrated skin screening approach, which is also recommended by the WHO.[1 19 20] Skin diseases are among the most common human illnesses, affecting almost 900 million people worldwide.[23] They are thought to be the fourth leading cause of global non-fatal disease burden and can result in disabilities, stigmatisation and discrimination.[23 97] Additionally, dermatological problems can be the first expression of systemic or chronic diseases, including HIV/AIDS, diabetes and NTDs.[17 98] Integrated skin screening is therefore expected to generate a greater health benefit compared with vertical health programmes which focus on one disease only. Pooling diseases in projects like PEP4LEP can also be helpful in educating and in raising awareness, as health workers' knowledge of NTDs like leprosy has been declining.[52 99 100] This was reflected in a study performed by Abeje et al among general health workers diagnosing leprosy in Ethiopia, which revealed that only 18% diagnosed leprosy correctly.[101] Detecting skin NTDs like leprosy therefore requires capacity-strengthening programmes.[17 19 25–29]

This study will also use mHealth solutions to support peripheral health workers in recognising and treating signs and symptoms of skin diseases. 'Digital health applications in leprosy' is described as key research topic in the WHO 'Global Leprosy Strategy 2021–2030'.[8] Evidence indicates that mobile technology tools can substantially benefit healthcare workers, their patients and adequate healthcare delivery.[102] In dermatology, electronic health (eHealth) was adopted early, with teledermatology as a widespread example, fostering the possibility of remote patient care and education.[103 104] This is especially valuable if health services are scarce and during periods of service disruption (eg, flooding, civil unrest, COVID-19 pandemic).[36 58 61 104 105] We emphasise the importance of studying the effect of mHealth technologies, aimed at capacity strengthening, like NLR's SkinApp, before fully focusing on upscaling.[30 31 61 102]

Despite the conclusion of the expert meeting that SDR-PEP poses negligible risk of generating rifampicin resistance in M. tuberculosis, ongoing resistance surveillance is important to consider.[15 106–108] However, because of the limited study period, resistance surveillance in the PEP4LEP implementation areas alone would add no value to the project as the number of patients will be too small and the project duration would be too short for any resistance to emerge during that period. It is therefore recommended to integrate the surveillance of rifampicin resistance in the PEP4LEP project areas with the resistance surveillance systems for TB and leprosy during the project period and beyond, consistent with WHO recommendations on resistance surveillance.[106–108]

Although SDR-PEP has been adopted in the WHO guidelines on leprosy, little is known about the feasibility of several implementation methods of SDR as chemoprophylaxis for leprosy in combination with varying and integrated contact screening methods, especially in sub-Saharan Africa.[4] Tanzania was the only sub-Sahara African country included in the LPEP programme.[13] Ortuno-Gutierrez et al recently outlined the Post-ExpOsure Prophylaxis for LEprosy in the Comoros and Madagascar (PEOPLE) study protocol.[109] PEOPLE assesses the effectiveness of different modalities of SDR-PEP, using door-to-door surveys and a double dosage of SDR-PEP. Both the PEOPLE and the PEP4LEP research questions comply with the Aligned Research Agenda for Zero Leprosy from the GPZL regarding the call for more operational studies and research focusing on SDR-PEP and on digital health.[110 111] Too often, innovative medical interventions fail because the factors contributing to success are poorly understood and hence not considered.[112] Lessons learned from SDR-PEP implementation are also expected to be relevant when improved preventive approaches, such as new chemotherapeutic regimens and vaccines, become available in the future.[8 108] Therefore, our goal is to share key insights gained from the PEP4LEP study to foster the implementation of integrated skin screening and chemoprophylaxis for leprosy in the sub-Sahara African context, which may also be relevant for the global leprosy community.

## Author affiliations

[1]Medical Technical Department, NLR, Amsterdam, The Netherlands
[2]Department of Public Health, Erasmus MC, University Medical Center Rotterdam, Rotterdam, The Netherlands
[3]Deutsche Lepra- und Tuberkulosehilfe e.V, Wuerzburg, Germany
[4]Armauer Hansen Research Institute, Addis Ababa, Ethiopia
[5]Lúrio University, Nampula, Mozambique
[6]Nampula Provincial Health Directorate, Ministry of Health Mozambique, Maputo, Mozambique
[7]Department of Microbiology and Immunology, Catholic University of Health and Allied Sciences, Mwanza, United Republic of Tanzania
[8]Ministry of Health Ethiopia, Addis Ababa, Ethiopia

⁹Ministry of Health Tanzania, Dodoma, United Republic of Tanzania
¹⁰Deutsche Lepra- und Tuberkulosehilfe e.V. Ethiopia, Addis Ababa, Ethiopia
¹¹NLR Mozambique, Nampula, Mozambique
¹²Deutsche Lepra- und Tuberkulosehilfe e.V. Tanzania, Dar es Salaam, United Republic of Tanzania

**Acknowledgements** Our thanks go to all those involved in the PEP4LEP project, including the study participants; the full research consortium; and our funders European and Developing Countries Clinical Trials Partnership (EDCTP) and Leprosy Research Initiative (LRI).

**Contributors** LM, CK, JHR, AS, TH and RvW designed the study. KB, FM, SEM, EM, AM, NM, TL, AM, VK, AME, LR and BN supported the development of country-specific protocols, materials and coordinate the study implementation. AS, TH and RvW have drafted the manuscript. All authors have reviewed the draft manuscript and have read and approved the final version.

**Funding** This project was supported by the EDCTP2 programme under Horizon 2020 (grant number RIA2017NIM-1839-PEP4LEP). The project also received funding from the Leprosy Research Initiative (LRI; www.leprosyresearch.org) under LRI grant number 707.19.58. Both funding bodies reviewed and approved the study proposal.

**Map disclaimer** The depiction of boundaries on the map(s) in this article does not imply the expression of any opinion whatsoever on the part of BMJ (or any member of its group) concerning the legal status of any country, territory, jurisdiction or area or of its authorities. The map(s) are provided without any warranty of any kind, either express or implied.

**Competing interests** None declared.

**Patient and public involvement** Patients and/or the public were involved in the design, or conduct, or reporting, or dissemination plans of this research. Refer to the Methods section for further details.

**Patient consent for publication** Not required.

**Provenance and peer review** Not commissioned; externally peer reviewed.

**ORCID iDs**
Anne Schoenmakers http://orcid.org/0000-0003-3040-7883
Thomas Hambridge http://orcid.org/0000-0003-2600-7510
Robin van Wijk http://orcid.org/0000-0002-0969-4155
Christa Kasang http://orcid.org/0000-0003-0241-0321
Jan Hendrik Richardus http://orcid.org/0000-0003-0564-6313
Kidist Bobosha http://orcid.org/0000-0001-8091-7182
Fernando Mitano http://orcid.org/0000-0003-4069-9314
Stephen E Mshana http://orcid.org/0000-0002-7526-6271
Ephrem Mamo http://orcid.org/0000-0003-1405-2568
Abdoulaye Marega http://orcid.org/0000-0002-7862-373X
Nelly Mwageni http://orcid.org/0000-0002-9204-7652
Artur Manuel Muloliwa http://orcid.org/0000-0003-2338-242X
Deus Vedastus Kamara http://orcid.org/0000-0002-2141-0696
Litos Raimundo http://orcid.org/0000-0002-1030-3406
Blasdus Njako http://orcid.org/0000-0002-8156-2579
Liesbeth Mieras http://orcid.org/0000-0001-6943-1712

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
