## [Reviewer comments · BMJ Open]

ARTICLE DETAILS

TITLE (PROVISIONAL)	The PEP4LEP study protocol: Integrated skin screening and SDR-PEP administration for leprosy prevention. Comparing the effectiveness and feasibility of a community-based intervention to a health center-based intervention in Ethiopia, Mozambique and Tanzania
AUTHORS	Schoenmakers, Anne; Hambridge, Thomas; van Wijk, Robin; Kasang, Christa; Richardus, Jan Hendrik; Bobosha, Kidist; Mitano, Fernando; Mshana, Stephen E.; Mamo, Ephrem; Marega, Abdoulaye; Mwageni, Nelly; Letta, Taye; Muloliwa, Artur; Kamara, Vedastus; Eman, Ahmed; Raimundo, Litos; Njako, Blasdus; Mieras, Liesbeth

VERSION 1 – REVIEW

REVIEWER	Ortuno-Gutierrez, Nimer Damien Foundation, Project
REVIEW RETURNED	08-May-2021

GENERAL COMMENTS	PEP4LEP is designed to provide evidence on the effectiveness and feasibility of SDR-PEP comparing a community-based intervention to a health center-based intervention, it address important operational questions for reaching eligible contacts and integrates capacity building using innovative digital technologies for the diagnosis of leprosy and other neglected tropical diseases. The results will guide countries on scaling up PEP and on the implementation of capacity building strategies while involving the community. Minor items for consideration:  - Case detection delay: is the primary outcome. Probably patient and health system delay will be also collected. If so, it would be good to including on the manuscript. Patient and health system delay issues can guide further improvements. - Sample size: consider adding the power and other items as standard deviation that illustrates that the study will provide relevant results. -Randomization: adding how many health facilities (health centers/posts) are included can help for illustrating the operationalization of the study. Format:  - Reference 1 (line 489) the link is not working and the year of publication is 2018.
---

REVIEWER	Steinnmann, Peter University of Basel
REVIEW RETURNED	17-May-2021

GENERAL COMMENTS	Schoenmakers and co-authors describe here the protocol of the
---

	PEP4LEP study comparing two approaches for leprosy contact tracing and chemoprophylaxis. The tracing of the close contacts of newly diagnosed leprosy patients and their screening for signs of leprosy is an established approach to achieve early detection of leprosy cases while chemoprophylaxis with single-dose rifampicin has only recently been recommended by WHO. Scaling up these interventions to population level is a big challenge for underfunded leprosy control programs and weak health systems so the evidence generated in the frame of these studies will be relevant far beyond the study countries.  - Introduction, line 81: the latest data should be cited, namely the 2019 update published in 2020. - Introduction, line 97 and throughout the publication: the final results of the LPEP study have been published so the final publications should be cited (e.g. also line 379). - Objectives, line 123: what is a “cross-functional collaboration”? - Objectives, line 125-6: edit sentence, e.g. “...to contribute to the interruption of M. leprae transmission by...” - Methods and analysis, line 144-5: what is a “high distribution” do you mean highly focal? - Community-based skin camp intervention, line 178: how are index patients diagnosed and identified as suitable starting points for an activity? - Health center-based intervention for household contacts, line 198: a household definition should be provided, and mentioned that household composition may vary considerably across study sites. The number of household members seems rather conservative considering the household realities in some of the study sites. - Outcomes, line 253: revise “to lead to more cases improved early” - Primary outcome measures, line 258: the choice of case detection delay as a primary outcome and basis for the sample size calculation is the main concern of this reviewer: the choice is neither justified in the manuscript nor is it evident why case detection delay might be differently affected by the two approaches – both approaches are expected to contribute to early detection, and the main difference is operational, not the speed with which detection happens (in both cases following the identification of an index patient). - Secondary outcome measures, line 267: “social” – do you mean “societal”? - Additional objectives, line 279: training is mentioned but not described while it will be one of the main aspects of the study. Please describe the approach to training. - Sample size, line 302: how will the baseline be determined? - Sample size, line 304: it is unclear how a delay difference of 3 months might be achieved if follow-up contact tracing and screening are to happen within 3 months of index patient diagnosis (household-based contact tracing – the delay for skin camps should also be indicated at the appropriate location). - Ethics, line 383: why was availability of an emergency allergy kit only recommended and not ensured? - Discussion, line 439: the evaluation of the SkinApp is not described, and in general there is barely any mention of the approach for social science/qualitative approaches (acceptability etc.). This should be added. - Discussion: the representativeness of the study settings should be critically discussed and which countries/regions are likely to profit from the results considering that the overwhelming majority of leprosy patients are not found in East Africa and health systems are organized quite differently in various regions.
--	--

VERSION 1 – AUTHOR RESPONSE

Reviewer: 1

Comments to the Author:

PEP4LEP is designed to provide evidence on the effectiveness and feasibility of SDR-PEP comparing a community-based intervention to a health center-based intervention, it address important operational questions for reaching eligible contacts and integrates capacity building using innovative digital technologies for the diagnosis of leprosy and other neglected tropical diseases. The results will guide countries on scaling up PEP and on the implementation of capacity building strategies while involving the community.

Minor items for consideration:

- Case detection delay: is the primary outcome. Probably patient and health system delay will be also collected. If so, it would be good to including on the manuscript. Patient and health system delay issues can guide further improvements.

We thank the reviewer for their feedback. This information is now included in Table 3.

- Sample size: consider adding the power and other items as standard deviation that illustrates that the study will provide relevant results.

Thank you for your comment. We expanded the sample size section of the manuscript, including details on the standard deviation, design effect and qualitative objectives of the study.

- Randomization: adding how many health facilities (health centers/posts) are included can help for illustrating the operationalization of the study.

Thank you for your comment. We have added the total number of health facilities in each country to the randomization section.

Format:

- Reference 1 (line 489) the link is not working and the year of publication is 2018.

Thank you for pointing this out, we have changed the reference.

Reviewer: 2

Comments to the Author:

Schoenmakers and co-authors describe here the protocol of the PEP4LEP study comparing two approaches for leprosy contact tracing and chemoprophylaxis. The tracing of the close contacts of newly diagnosed leprosy patients and their screening for signs of leprosy is an established approach

to achieve early detection of leprosy cases while chemoprophylaxis with single-dose rifampicin has only recently been recommended by WHO. Scaling up these interventions to population level is a big challenge for underfunded leprosy control programs and weak health systems so the evidence generated in the frame of these studies will be relevant far beyond the study countries.

- Introduction, line 81: the latest data should be cited, namely the 2019 update published in 2020.

We thank the reviewer for their feedback and have updated the annual newly detected patient rate, year & reference.

- Introduction, line 97 and throughout the publication: the final results of the LPEP study have been published so the final publications should be cited (e.g. also line 379).

Thank you for highlighting this important point. This has been changed throughout the document.

- Objectives, line 123: what is a “cross-functional collaboration”?

Thank you for your comment. The word ‘cross-functional’ has been deleted.

- Objectives, line 125-6: edit sentence, e.g. “...to contribute to the interruption of *M. leprae* transmission by...”

Your suggestion was much appreciated, we changed the text accordingly.

- Methods and analysis, line 144-5: what is a “high distribution” do you mean highly focal?

Thank you for your comment, the sentence has been changed.

- Community-based skin camp intervention, line 178: how are index patients diagnosed and identified as suitable starting points for an activity?

Thank you for your comment. We added a referral to Table 1 and included a more detailed explanation on this point in the “Participants and eligibility criteria” section.

- Health center-based intervention for household contacts, line 198: a household definition should be provided, and mentioned that household composition may vary considerably across study sites. The number of household members seems rather conservative considering the household realities in some of the study sites.

Thank you for your comment. “Household contacts are defined as living under the same roof as the leprosy index patient for a minimum of three months” was included, we added a referral to Table 1.

- Outcomes, line 253: revise “to lead to more cases improved early”

Thank you for this remark, we have revised the sentence.

- Primary outcome measures, line 258: the choice of case detection delay as a primary outcome and basis for the sample size calculation is the main concern of this reviewer: the choice is neither justified in the manuscript nor is it evident why case detection delay might be differently affected by the two approaches – both approaches are expected to contribute to early detection, and the main difference

is operational, not the speed with which detection happens (in both cases following the identification of an index patient).

Thank you for highlighting this. Since the epidemiological impact (i.e. reduction in new case detection rate in PEP4LEP districts) will not become apparent within the study duration due to the long incubation time of leprosy, we selected case detection delay as a primary outcome measures of effectiveness in the comparison of the two interventions (and for the sample size calculation). We hypothesise that enhanced case finding and integrated skin screening will lead to an overall reduction of detection delay in the community-based intervention over the study duration, driven by diagnosis of patients with early signs of leprosy (and shorter delays) that would otherwise go undetected. We also agree both approaches are expected to contribute to early detection (compared to baseline), although we anticipate the largest reduction will be seen in the community-based intervention with more newly detected patients found during skin camp screening, presenting with shorter delays overall. The outcomes, sample size and strengths and limitations sections of the manuscript have been expanded to include this information.

- Secondary outcome measures, line 267: “social” – do you mean “societal”?

Thank you for noticing this, we agree with the point raised and have revised the sentence and removed the word ‘social’.

- Additional objectives, line 279: training is mentioned but not described while it will be one of the main aspects of the study. Please describe the approach to training.

Thank you for this great point, we included information on the training content in the methods section.

- Sample size, line 302: how will the baseline be determined?

Thank you for raising this point. A baseline case detection delay will be estimated in each country by interviewing recently diagnosed leprosy patients with the same structured questionnaire prior to the start of the study. This information has been added to the methodology section.

- Sample size, line 304: it is unclear how a delay difference of 3 months might be achieved if follow-up contact tracing and screening are to happen within 3 months of index patient diagnosis (household-based contact tracing – the delay for skin camps should also be indicated at the appropriate location).

Thank you for your comment. The sample size was calculated with conservative parameter assumptions to be able to detect a minimal difference in case detection delay of three months, although we expect the actual difference to be six months or more (baseline mean case detection delay is high in all three countries). As stated above, we hypothesise that the average detection delay for patients diagnosed in the skin camps will be lower overall due to widespread screening of community contacts, many of whom will present with very early clinical signs of leprosy.

- Ethics, line 383: why was availability of an emergency allergy kit only recommended and not ensured?

Thank you for raising this point. We changed the text and referred to our SOP and the national guidelines on this. References to these guidelines were also included.

- Discussion, line 439: the evaluation of the SkinApp is not described, and in general there is barely any mention of the approach for social science/qualitative approaches (acceptability etc.). This should be added.

Thank you for raising this point. More information on the qualitative component of the study was added in the methods section, as well as a reference to the more extensive (country-specific) protocols which will be written on this topic.

- Discussion: the representativeness of the study settings should be critically discussed and which countries/regions are likely to profit from the results considering that the overwhelming majority of leprosy patients are not found in East Africa and health systems are organized quite differently in various regions.

This point is very much appreciated, We included a dissemination section and also further addressed this point in the discussion.

VERSION 2 – REVIEW

REVIEWER	Ortuno-Gutierrez, Nimer Damien Foundation, Project
REVIEW RETURNED	04-Jul-2021
GENERAL COMMENTS	The authors included important elements on the background and methods that improved substantially the scope of the research for implementing innovative case finding using mobile technology and involving the community.